# Beyond the Flavour: The Potential Druggability of Chemosensory G Protein-Coupled Receptors

**DOI:** 10.3390/ijms20061402

**Published:** 2019-03-20

**Authors:** Antonella Di Pizio, Maik Behrens, Dietmar Krautwurst

**Affiliations:** Leibniz-Institute for Food Systems Biology at the Technical University of Munich, Freising, 85354, Germany; m.behrens.leibniz-lsb@tum.de (M.B.); d.krautwurst.leibniz-lsb@tum.de (D.K.)

**Keywords:** smell, taste, flavour molecules, drugs, chemosensory receptors, ecnomotopic expression

## Abstract

G protein-coupled receptors (GPCRs) belong to the largest class of drug targets. Approximately half of the members of the human GPCR superfamily are chemosensory receptors, including odorant receptors (ORs), trace amine-associated receptors (TAARs), bitter taste receptors (TAS2Rs), sweet and umami taste receptors (TAS1Rs). Interestingly, these chemosensory GPCRs (csGPCRs) are expressed in several tissues of the body where they are supposed to play a role in biological functions other than chemosensation. Despite their abundance and physiological/pathological relevance, the druggability of csGPCRs has been suggested but not fully characterized. Here, we aim to explore the potential of targeting csGPCRs to treat diseases by reviewing the current knowledge of csGPCRs expressed throughout the body and by analysing the chemical space and the drug-likeness of flavour molecules.

## 1. Introduction

Thirty-five percent of approved drugs act by modulating G protein-coupled receptors (GPCRs) [1,2]. GPCRs, also named 7-transmembrane (7TM) receptors, based on their canonical structure, are the largest family of membrane receptors in the human genome. The most commonly-used classification system divides GPCRs into six classes: class A (rhodopsin-like), consisting of over 80% of all GPCRs, class B (secretin-like), class C (metabotropic glutamate receptors), class D (pheromone receptors), class E (cAMP receptors) and class F (frizzled/smoothened family) [3,4,5].

The high GPCRs’ “druggability”, that is, the likelihood of modulating a target by small-molecule drugs [6], is due to a combination of numerous factors, including their physiological and pathological relevance, their expression in the plasma membrane, which facilitates molecular interactions in the extracellular milieu, and a very defined binding site [2,7,8,9]. The number of GPCRs targeted by drugs is currently 134, ~16% of the ~800 GPCRs in the human genome [1]. Remaining 84% of the GPCR repertoire include orphan GPCRs (~100 receptors) and sensory GPCRs (olfactory, taste and visual receptors). Chemosensory GPCRs (csGPCRs) constitute structurally and phylogenetically diverse subgroups within the superfamily of GPCRs (Figure 1).

Odorant receptors (ORs) are class A GPCRs, encoded by ~400 genes in human and thus represent the largest GPCR subgroup. Additional class A csGPCRs in the nasal cavity are the trace amine-associated receptors (TAARs). Six functional *TAAR* genes have been found in human and except for TAAR1, all receptor subtypes are suggested to be targeted by odorants, particularly by volatile amines [10,11,12].

Bitter taste signalling is initiated by 25 TAS2Rs [13,14,15], classified as class A GPCRs for their architecture and binding site location [16]. By contrast, there are only three class C GPCRs, TAS1R1, TAS1R2 and TAS1R3, which form functional heterodimers that specifically recognize sweeteners and amino acids: the TAS1R2/TAS1R3 combination recognizes natural and artificial sweeteners whereas the TAS1R1/TAS1R3 is involved in umami taste [17,18,19,20]. GPCRs have been suggested to mediate also the orosensory perception of fat [21,22,23] and kokumi (i.e., enhancement of mouthfulness and thickness of food perception) substances [24,25].

It is well-established that, by csGPCRs within our chemical senses smell and taste, we constantly monitor our external chemical environment to detect and discriminate especially foodborne stimuli [26,27]. However, canonical csGPCRs are also expressed in tissues not directly related to the detection of odorants or tastants [28,29,30,31], which strongly suggest their role in the monitoring of internal environments. Indeed, in some cases their physiological roles and their involvement in serious diseases, like respiratory and metabolic diseases and even cancer, have been characterized [32].

The fact that csGPCRs have pharmacological relevance and that they belong to a highly druggable protein family, unequivocally points them out as new attractive drug targets, increasing the potential therapeutic GPCR target space [33]. Since previous analyses on emerging drug design trends and opportunities have focused on the non-sensory GPCR-ome [1,2,34,35], in this review we attempt to describe the state-of-the-art of csGPCR research and explore the potential druggability of csGPCRs.

## 2. “Ecnomotopic” csGPCRs and Their Modulation by Small Molecules

*OR* gene expression has been established in a multitude of human tissues, that is, brain, blood leukocytes, airway smooth muscle, skin, gut [36,37,38,39,40,41,42,43,44,45]. Similarly, TAS1Rs and TAS2Rs are located in tissues other than the tongue and palate epithelium; including gastrointestinal tract, heart, leukocytes, vascular smooth muscle, airway epithelium, skin, lung and brain [39,46,47,48,49,50,51,52,53,54,55].

The expression of extra-nasal ORs and extra-oral TAS1Rs and TAS2Rs is frequently defined as “ectopic” [etymology: from Greek, ἐκ(out) + τόπος(place), *out-of-the-place*] [32,56]. Ectopic is a medical term used for “a biological event or process occurring in an abnormal location or position within the body” [56,57]. However, csGPCRs outside their canonical places cannot be considered abnormal and we prefer to use the term “ec-nomotopic” [etymology: from Greek, ἐκ(out) + νόμος(law, custom) + τόπος(place), *out-of-the-usual(conventional)-place*]. To the best of our knowledge, the term nomotopic is used in medicine for biological events “occurring at the usual place” while the term ecnomotopic can be considered as a new combined word [57]. Indeed, we are approaching a unique case: receptors that are named as taste and smell receptors but their function might be not limited to the taste and smell perception.

The number of csGPCRs expressed in different human tissues varies substantially, some have a broad tissue distribution, whereas others appear to be exclusively restricted to one specific tissue [37,58]. Overall, csGPCRs have a tissue-dependent expression, with generally lower average expression level in ecnomotopic tissues compared to that observed in the respective sensory tissues [59,60].

Even though the biological functions of ecnomotopic csGPCRs have not been fully characterized, they seem to have the potential to serve as therapeutic tools [61,62,63,64,65,66]. The current knowledge about the ecnomotopic expression of smell and taste receptors and their suggested physiological and pathological functions has been recently and carefully reviewed [29,32,67]. Here, we aim to lay the bases for analysing and discussing the potential use of drug design techniques for treating diseases through the chemosensory reception system; therefore, we zoom-in on those cases where the putative biological functions of ecnomotopic csGPCRs were tested and reversed with the use of small-molecule ligands.

### 2.1. Ecnomotopic ORs

Very recently, OR2AT4 expressed in human scalp hair follicles has been found as a target for hair loss therapy: indeed, stimulation of OR2AT4 by the odorant Sandalore (3-methyl-5-(2,2,3-trimethylcyclopent-3-en-1-yl)pentan-2-ol) prolongs human hair growth by decreasing apoptosis and increasing production of the anagen-prolonging growth factor IGF-1; in contrast, co-administration of the specific OR2AT4 antagonist (Phenirat, 2-phenoxyethyl 2-methylpropanoate) inhibits hair growth [68]. Preliminary studies for the use of Sandalore in shampoo or lotion have been performed and clinical trials are planned in Italy.

Some ORs are highly expressed in cancer tissues and this opened new directions for cancer diagnosis [69,70,71,72,73,74,75]. Additionally, in some specific cases, the role of ORs in cancer tissues has been characterized, paving the way to target ORs as a strategy of cancer therapy. OR51E2, one of the most broadly expressed ecnomotopic ORs, is present in healthy prostate tissue and shows significantly increased expression in prostate adenocarcinoma [37,75]. There are several lines of evidence that the OR51E2 agonist β-ionone has effects on prostate cancer: it can cause a decreased proliferation but also an increase of the invasiveness of human prostate cancer cells [76,77,78]. Recently, a testosterone metabolite (19-hydroxyandrostenedione) was found to be an endogenous agonist produced by activation of OR51E2 in prostate cancer cells [79]. The activation of OR51E2 by newly endogenous metabolites induces neuroendocrine trans-differentiation, a feature commonly seen in prostate carcinoma and frequently associated to hormone therapy refractory and aggressive disease [79,80].

Stimulation of another OR expressed in prostate tissue, OR51E1, with its agonist nonanoic acid was found to significantly reduce proliferation and induce cellular senescence in prostate cancer cell line LNCaP and to influence androgen receptor-mediated signalling [81].

Activation of OR2J3 in human lung carcinoma tissues with the agonist helional (3-(1,3-benzodioxol-5-yl)-2-methylpropanal) was found to induce apoptosis and inhibition of cell proliferation [82]. Apoptosis and inhibition of cell proliferation are also caused by stimulation of OR51B4 by Troenan (5-methyl-2-pentan-2-yl-5-propyl-1,3-dioxane) in colorectal cancer cells [83]. Suppository capsules with Troenan for colon cancer have not been tested in clinical trials but are already being used to treat patients in German clinics [32].

### 2.2. Ecnomotopic Taste Receptors

Accumulating evidence supports the role of csGPCRs in the gut for monitoring foodborne compounds: enteroendocrine cells express csGPCRs, which sense food components and metabolites and by regulating hormones such as glucagon-like peptide 1 (GLP-1) and ghrelin transmit signals to control the secretion of appetite [42,55,84,85].

Stimulation of sweet taste receptors in the duodenal L cells by glucose and sucralose was shown to cause the release of GLP-1 that can be reversed by the sweet receptor antagonist lactisole [86,87]. GLP-1 increases insulin release from beta cells and inhibits glucagon release and gastric emptying. Therefore, modulating GLP-1 secretion in gut via sweet taste receptors may provide an important treatment for obesity, diabetes and abnormal gut motility [86]. However, in-vivo studies on the effect of artificial sweeteners on GLP-1 secretion gave controversial results: the oral delivery of sweet tastants seems not to be associated with GLP-1 release but the increase in GLP-1 secretion can still be observed for combinations of artificial sweeteners in diet drinks [88,89,90,91].

The activation of TAS2R9 in cells of enteroendocrine origin was found to elicit GLP-1 secretion [92]. Similarly, mouse gut-expressed bitter taste receptors stimulated with denatonium benzoate lead to GLP-1 secretion [93]. The anti-diabetic properties of the bitter isohumulone KDT501, on phase II clinical trial (ID number: NCT02444910) for threating metabolic disorders, were recently found to be mediated by bitter taste receptor Tas2r108 in mice (and supposedly by TAS2R1 in humans, since KDT501 can activate selectively this receptor) [94].

Intragastric administration of the bitter denatonium benzoate in humans (healthy females) was shown to decrease hunger scores and caloric intake [95]. Intragastric administration of denatonium benzoate in mice can induce release of ghrelin and, consequently, decrease food intake [96].

On the contrary, intragastric administration of glucose was found to reduce plasma ghrelin levels [97]. Recently, Wang and colleagues have performed in-vitro studies to show how the chemosensory signalling in the gut is altered in obesity: in fundic cultures of obese patients, glucose inhibited ghrelin secretion via TAS1R3 and bitter compounds, such as chloroquine, stimulated ghrelin secretion [84].

Ecnomotopic taste receptors seem to work together also in the upper airway. A robust innate immune defensive function of the upper airway, the production of nitric oxide, is mediated by TAS2R38 in response to acyl-homoserine lactones, that is, N-butanoyl-L-homoserine lactone (C4HSL) and N-lauroyl-L-homoserine lactone (C12HSL), quorum sensing molecules secreted by bacteria [98,99,100]. Similarly, nitric oxide production can be induced by TAS2R14 activation with flavones (e.g., apigenin and chrysin) and can be blocked with TAS2R14 antagonists (e.g., 4′-fluoro-6-methoxyflavanone) [101]. The TAS2R-mediated anti-microbial activity was found to be inhibited by the sweet taste receptor [102,103]. TAS1R2/TAS1R3 was suggested also to measure glucose concentration in the mucosa: as microorganisms use the sugar in their environment for energy consumption, a drop in glucose concentration would indicate infection, which would in turn remove the sweet taste roadblock to activate bitter taste-induced immunity [102,103].

Bitter taste receptors are then suggested as responsible for the anti-inflammatory activity observed in asthma, since the stimulation of TAS2Rs in mast cells with bitter agonists (i.e., chloroquine and denatonium) mediate the inhibition of an IgE-induced release of histamine [104].

Stimulation of TAS1R3 in the pulmonary vasculature by the artificial sweetener sucralose was demonstrated to affect pulmonary microvasculature and may represent a novel therapeutic strategy to protect the endothelium in settings of acute respiratory distress syndrome [52].

### 2.3. Ecnomotopic Odorant and Taste GPCRs: A Functional Cross-Talk?

The expression of different csGPCRs in same ecnomotopic tissues, such as the case of sweet and bitter taste receptors in the gut and the upper airway, seems not to be exceptional and not even confined to taste modalities.

One of the most relevant example is the presence of csGPCRs in the leukocytes: TAARs, ORs, TAS2Rs, sweet and umami taste receptors are all expressed in leukocytes and investigating their roles is a growing research field [45,51,105,106]. The cellular immune system is exposed to exogenous foodborne chemicals and may respond to them through csGPCRs. In fact, OR56B4 in human leukocytes, upon agonist activation, for example, with δ-decalactone, induces concentration-dependent chemotaxis of isolated human neutrophils [107].

Both olfactory and bitter taste GPCRs are expressed in the human airway smooth muscle (ASM). Bitterants, such as chloroquine and quinine, can act as bronchodilator by stimulating TAS2Rs in the ASM [108,109,110], whereas OR51E2, activated by its agonists, such as acetate and propionate, causes a decrease of cytoskeletal remodelling and proliferation, two cardinal features of asthma [111].

To date, most studies on ecnomotopic csGPCRs have focused on gene expression rather than on their physiological relevance, therefore we do not have proofs to support the complementary/synergistic functions of different csGPCRs when expressed in the same tissues. Certainly, the high physiological and pathological potential on one side and the complexity of the cross-talking between the csGPCRs on the other side, suggest that it would be of great value to use csGPCR modulators to efficiently shed light on their biological functions and attempt to treat csGPCR-mediated pathological disorders.

Figure 2 shows the chemical structures of the csGPCRs ligands mentioned in this paragraph. Can these compounds be considered candidates or starting points for the development of csGPCR drugs?

## 3. csGPCR Drug Discovery

Drug discovery is a multi-step process that starts with target identification (Figure 3). Thinking of filling in a hypothetical drug discovery pipeline for csGPCRs, ecnomotopic odorant and taste receptors described in the paragraph above can be considered as the identified biological targets. The next steps, hit discovery and optimization and lead development, aim to provide good candidates for preclinical and clinical studies.

### 3.1. csGPCR Hit Discovery

Known ligands of chemosensory receptors mainly come from the flavour chemistry, synthetic or natural compounds with well-defined sensory properties. The recently developed “sensomics” approach has led to the identification of many taste and odorant molecules from complex processed foods [112,113,114,115,116,117,118]. Several csGPCRs have been deorphanized with in-vitro screening of odour- and taste-active compounds [26,119,120,121,122,123,124,125,126,127].

Sweet taste molecules include monosaccharides (e.g., glucose, fructose, galactose), disaccharides (e.g., sucrose, lactose, trehalose), polyols (e.g., sorbitol, mannitol, xylitol), D-amino acids (e.g., D-Tryptophane), proteins (e.g., thaumatin), plant-derived sweeteners (e.g., neohesperidin dihydrochalcone, stevioside and steviolbioside) and synthetic non-nutritive sweeteners (e.g., saccharin, sucralose, aspartame) [128,129]. Umami molecules are proteogenic amino acids or peptides, but also small molecules [129,130,131]. Bitter molecules include ions, peptides, alkaloids, polyphenols and glucosinolates [132,133]. A comprehensive list of plant naturals as agonists/antagonists for taste receptors has been recently reviewed [134]. OR agonists are volatile compounds, of less than 400 Daltons [27].

Can these flavour compounds be considered hits for drug discovery? Are they good candidates for further optimization?

Hit molecules are chemical compounds capable to interact with the target with micromolar potency. Many csGPCRs are still orphan and their ligands need to be discovered. For deorphanized csGPCRs, most identified csGPCR ligands are agonists with mid-to-high micromolar potency, a range that is relevant for flavour detection but may not be sufficient for ecnomotopic functions and is definitely not enough for drug discovery [27,135]. Therefore, to be considered proper hits their potency should be optimized.

Importantly, because of the high cost and low success rate of the drug discovery process, promising drug candidates should be selected at early stages to avoid late failures. In pharma terms, it is important that hit molecules are “drug-like” [136]. Drug-like compounds have molecular properties consistent with the majority of known drugs and hence can be inferred as compounds with therapeutic potential [137]. Drug-likeness provides a broad composite descriptor that implicitly captures several criteria, with bioavailability among the most prominent. Methods that predict drug-likeness of molecules are based on their ability to distinguish known drugs from non-drugs [138]. Interestingly, properties relevant for drug delivery, like solubility, metabolic stability, oral bioavailability, membrane permeability, are often correlated with molecular descriptors. Several methods have been developed to predict the range of properties where drug-like molecules fall, from simple counting of descriptors to methods that take into consideration chemistry or structural features [138,139,140,141,142,143,144,145].

Simple counting methods include Lipinski’s RULE OF 5 (RO5) [144]. The RO5 states that molecules are more likely orally bioactive if they have hydrogen-bond donors (HBDs) fewer than 5, hydrogen-bond acceptors (HBAs) fewer than 10, Molecular Weight (MW) not over 500 Daltons, calculated octanol–water partition coefficient (log P) not greater than 5. Even though passing the RO5 is no guarantee that a compound is drug-like, because of its simple and straightforward applicability, it is a widespread guideline for compound evaluation. And, indeed, the preferential selection of drug-like compounds helps alleviate attrition rates in drug discovery [146].

DrugBank is a freely accessible database containing information on approved drugs and drug candidates [147,148]. Since 1997, when Lipinski’s rule was defined, to our days, the chemical space of drugs has seen a substantial growth. The latest release of DrugBank (version 5.1.1) contains more than 10,000 drug entries including ~2500 approved drugs. RO5 applies to ~70% of approved drugs and ~60% of all DrugBank entries. The majority of approved drugs has MW between 200 and 500 Daltons (66%), HBAs fewer than 5 (80%), HBDs fewer than 3 (88%), a wider distribution of AlogP, with 75% of drugs ranging from −1 to 5 (Figure 4).

### 3.2. Drug-Likeness of Flavor Molecules

Flavour molecules shown in Figure 2 have high chemical diversity: different hydrophobic content, few HB donors and acceptors and low MW, reaching a maximum of 365 for the bitter isohumulone. For a better analysis of the chemical space, we analysed molecular properties of cured libraries of flavour molecules. The new release of BitterDB stores more than 1000 bitter molecules, including calculated physicochemical properties [149,150]; 100 sweet tastants (Appendix A) were collected by integrating small molecules listed in the SuperSweet database [151] with natural compounds reviewed by Kim and Kinghorn in 2002 and Behrens and colleagues in 2011 [129,152]; 37 umami compounds (Appendix A) were retrieved from literature [131,153,154]; and the in-house list of odorant molecules characterized in food samples was made available in its updated version (250 compounds instead of the 226 molecules published in 2014) [27]. Table 1 summarizes the analysis of physicochemical properties and drug-likeness calculated for the collected flavour molecules. We consider here drug-like those molecules that have MW between 200 and 500 Daltons, AlogP between −1 and 5, HBAs fewer than 5 and HBDs fewer than 3, that is, the ranges were most of approved DrugBank compounds fall (Figure 4).

The shared chemical space between drugs and bitter compounds has been previously investigated [132,155]: many drugs suffer low compliance for their bitter taste and almost 10% BitterDB compounds are approved drugs. The numbers in Table 1 are very promising and the majority of bitter compounds can be considered drug-like. Interestingly, compared to other flavour sets, bitter compounds show a wider range of AlogP. This leads us to think that bitter compounds carry higher chemical diversity, however we have to consider the diversity in set composition (1000 bitter compounds are analysed compared to less than 500 for all other flavours). The connection between bitterness and hydrophobicity has been extensively disputed [132,133,156]. Even though hydrophobicity alone was not found to be a predictive feature of small molecules’ bitterness [132], the majority of bitter compounds shows higher hydrophobicity compared to sweet compounds and reducing hydrophobicity has been suggested as a strategy for decreasing the bitter off-taste of sweeteners [128]. Odorant molecules are very small molecules and consequently the ranges of their properties are tinier, making this set more difficult to compare to other sets but definitely the less drug-like (Table 1). However, the OR51E2 endogenous agonist 19-hydroxyandrostenedione reported in Figure 2 suggests that OR endogenous ligands may cover a different chemical space compared to odorants and may be good candidates for OR drug design.

Despite the chemical diversity, a shared chemical space between csGPCR ligands can be envisioned. Indeed, many sweeteners are limited in their use due to the unpleasant bitter off-taste [157,158]. There are many examples of sweet compounds that can modulate bitter taste receptors [128]. TAS2R4 and TAS2R14 were found to mediate the bitter off-taste of steviol glycosides [159]. The sweet saccharin activates TAS2R31 and TAS2R43 and blocks TAS2R1 and TAS2R38 [160,161]. On the contrary, cyclamate is a sweetener that activates TAS2R1 and TAS2R38 and blocks TAS2R31 and TAS2R43 [160]. The odorant (R)-citronellal has been recently reported as an inhibitor of the bitter taste receptors TAS2R43 and TAS2R46, indicating a possible shared chemical space between OR and TAS2R ligands [162]. Exploring the overlapping chemical space of csGPCR ligands may contribute to decode the complex receptor-based mechanisms underlying chemosensation but also shed light on the ecnomotopic csGPCRs’ molecular pharmacology.

### 3.3. Towards the Discovery of csGPCR Lead Molecules

During the H2L stage, the chemical space around each hit is explored in order to narrow down the candidate molecules to drug-like high-potency lead structures, that is, molecules that potently trigger or antagonize an intrinsic activity of the target, while confining logP and MW, in order to penalize compounds that improve potency with unnecessary increases in molecular size and/or lipophilicity [146,163,164].

Bitter drugs could be located at this stage of our drug discovery pipeline (Figure 3). Even though these molecules are already approved drugs, their potency against TAS2Rs is not enough for their repurposing to TAS2R-driven diseases [135]. Flufenamic acid is an anti-inflammatory and analgesic drug and, together with strychnine [165], is the most potent known TAS2R agonist but its activity towards its cognate bitter taste receptor, TAS2R14, requires concentrations of 100 nM or higher [119].

Ligand- and structure-based computer-aided drug design (CADD) tools are commonly used in this drug discovery stage to improve binding, selectivity or other pharmacological properties of hit molecules [166,167]. To extend the chemical space of hit series, analogues can be quickly selected from commercially available compound libraries (“analogue by catalogue”) and/or generated with “hit expansion” tools. Testing of analogue compounds will provide data for ligand-based analysis to determine a quantitative structure-activity relationship (QSAR). Structure-based approaches can be used to investigate the key interactions between the receptor and the ligand for the rational design of new molecules with improved binding and selectivity profile.

As we discussed in the previous paragraph, taste and odorant compounds are high-micromolar csGPCR agonists. Until now, the use of CAAD methods to csGPCRs has been confined to hit identification and rationalization of known data [16,168,169]. Ligand-based methods were used to investigate the activity and selectivity of flavonoids and isoflavonoids towards TAS2R14 and TAS2R39 [170]; to identify TAS2R14 agonists among approved and experimental drugs [171]; to identify bitter taste antagonists [172,173], to discriminate agonist vs. antagonist molecular determinants towards OR1G1 [174], to capture the key features for the interaction between umami compounds and its receptor [153].

Figure 1 shows the schematic representation of the different classes of csGPCRs. Despite the great progress in GPCR structure determination, no experimental structure of a csGPCR is currently available and 3D structures of csGPCRs are computationally modelled [16,175]. The most used technique, homology modelling, predicts protein structure starting from the structure(s) of homologous protein(s), therefore the quality of the generated model is determined by the template selection [176]. Several class A and class C GPCRs have been crystallized but the similarity with csGPCRs is low and leads to low-resolution models. Despite that, the analysis of a wide conformational space of the binding site and the integration with experimental data allowed the validation of the structure predictions and the investigation of the ligand binding modes [177,178].

With the contribution of structure-based methods, it was found that the umami receptor orthosteric ligand binding site is located in the extracellular Venus flytrap (VFT) domain of the TAS1R1 subunit and the sweet receptor orthosteric ligand binding site in the VFT domain of the TAS1R2 subunit [179,180,181,182]. The predicted binding modes of known umami and sweet taste enhancers have even opened the possibility to rationally design potent allosteric modulators for class C csGPCRs [181,182,183,184,185].

By combining CADD methods with site-directed mutagenesis and functional analysis, a descriptive and predictive molecular model was generated for the OR2AG1 receptor and its agonists [186]. Integrated computational/experimental approaches were recently used to characterize the enantiomer-selective carvone binding pocket of the OR1A1 receptor [187] and the binding mode of muscone analogues into the OR5AN1 and OR1A1 receptors [188]. A virtual screening campaign against the 3D homology model of OR51E2 successfully led to the identification of potent ligands among human metabolite compounds [79]. Homology modelling and virtual screening techniques were also used to discover new ligands for mouse TAAR5 receptor [189].

Despite the low similarity of TAS2Rs to class A GPCRs (less than 30% for the TM domains), the combination of structure-based CADD approaches and in-vitro techniques allowed to locate the ligand binding pocket of bitter taste receptors in the extracellular side of the TM bundle, as the well-characterized orthosteric binding pocket of class A GPCRs [135]. The multi-specificity of TAS2Rs towards a vast range of chemical structures seems to be achieved by using subsites within the binding pocket and by forming different types of interactions for different ligands [190,191,192,193,194,195,196]. Interestingly, it has been suggested that, similarly to class A GPCRs, TAS2Rs possess an additional vestibular binding site transiently occupied by agonists [197]. The first attempts of using CADD methods for hit optimization of csGPCR ligands have been very recently pursued: hit expansion of the selective TAS2R14 agonist flufenamic acid has led to new agonists with improved potency compared to the reference structure [198].

## 4. Conclusions

More than half of the GPCR family members are chemosensory receptors, involved in taste and olfaction, along with a variety of other physiological processes when expressed ecnomotopically. Hence, these receptors constitute promising targets for pharmaceutical intervention.

Until now, chemosensory research has focused on flavour molecules, indicative of food content, as csGPCR ligands: taste molecules, as the attractive sweet and umami (indicative of carbohydrate and protein content, respectively) and the aversive bitter (indicative of toxicity) and odorant molecules, volatile compounds responsible for food aroma. As first measurement of food quality, these molecules activate csGPCRs at relatively high concentrations. This is a limit also for the repurposing of bitter drugs, since most compounds currently tested in-vitro against TAS2Rs showed at most sub-micromolar activity.

Flavour molecules are chemically diverse: odorants are usually smaller than tastants, bitter compounds tend to be more hydrophobic than sweet and umami compounds. Bitter molecules are the most and the odorants the least drug-like flavour compounds. The chemical space and biological efficacy of flavour molecules as csGPCR ligands limit their current use for therapeutic applications. CADD methods, already in use for understanding of csGPCR molecular recognition, are suggested as useful tools to speed up the process of hit identification and optimization for ecnomotopic csGPCRs.

## Figures and Tables

**Figure 1 ijms-20-01402-f001:**
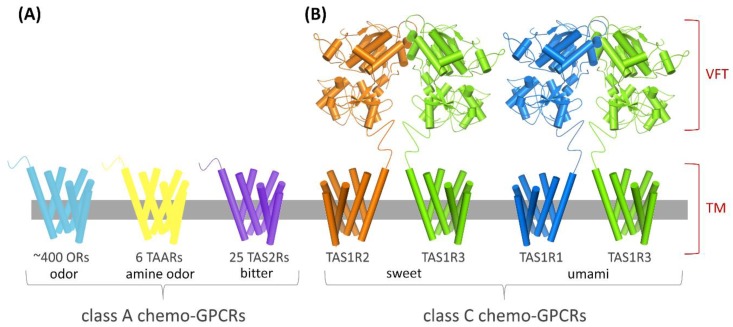
Architecture of csGPCRs. (**A**) Chemosensory receptors classified as class A G protein-coupled receptors (GPCRs), whose orthosteric binding site is located inside the TM domain, i.e., ORs, TAARs and TAS2Rs. (**B**) Chemosensory receptors classified as class C GPCRs, whose orthosteric binding site is located in the Venus flytrap (VFT) domain, i.e., TAS1R2/TAS1R3 and TAS1R1/TAS1R3.

**Figure 2 ijms-20-01402-f002:**
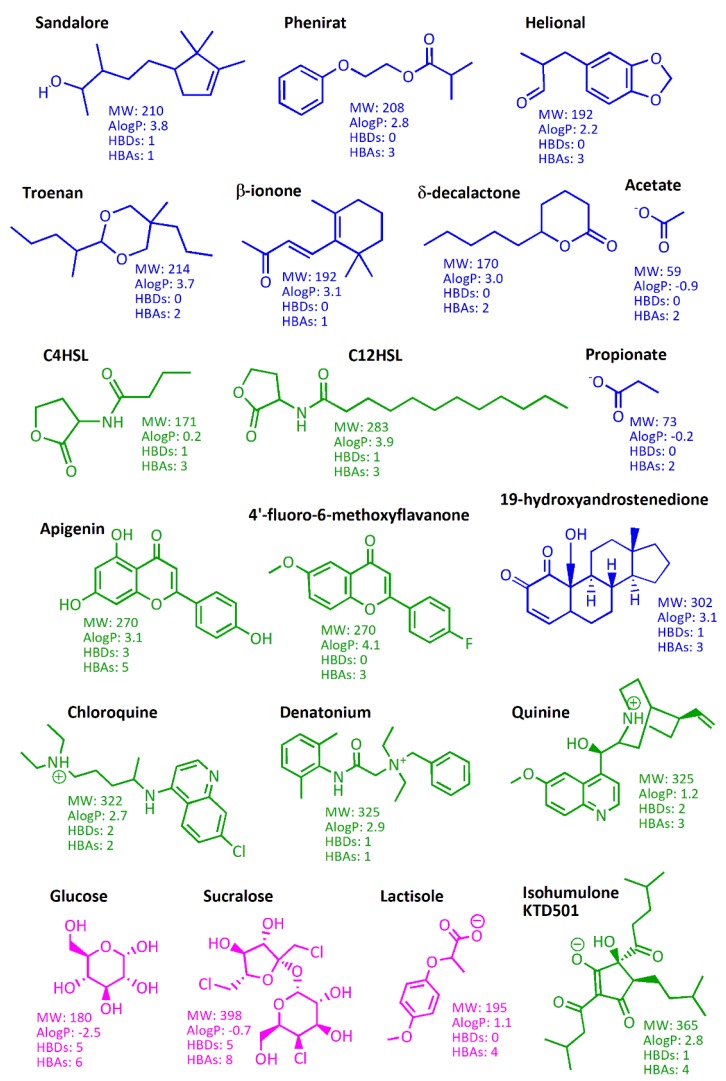
Chemical structures and physicochemical properties of csGPCR ligands. **Odorant receptor** (OR), TAS2R and TAS1R ligands are coloured in blue, green and magenta, respectively. MW (Molecular Weight), AlogP (octanol/water partition coefficient), HBDs (Hydrogen Bond Donors) and HBAs (Hydrogen Bond Acceptors) values at physiological pH have been calculated with Maestro (Schrödinger Release 2018-4: Maestro, Schrödinger, LLC, New York, NY, 2018).

**Figure 3 ijms-20-01402-f003:**
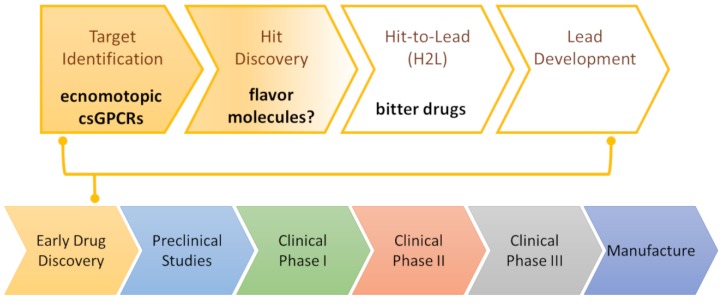
Drug discovery pipeline for csGPCRs.

**Figure 4 ijms-20-01402-f004:**
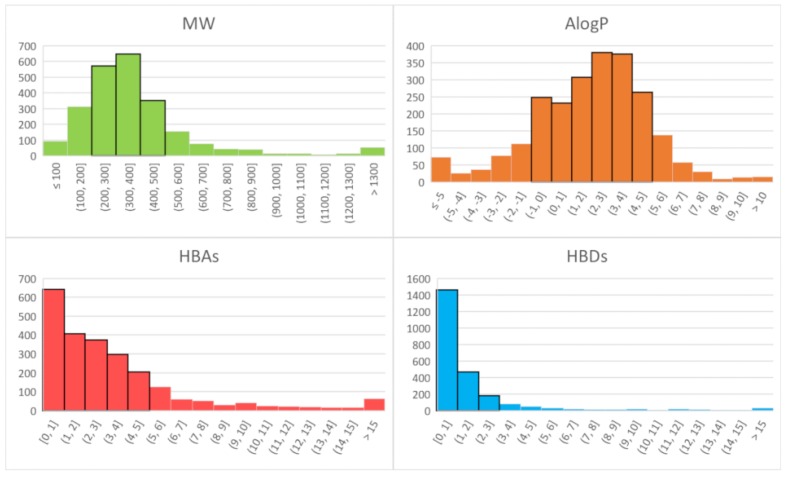
Distribution of molecular descriptors defining the RO5 among approved drugs (DrugBank v. 5.1.1). MW, AlogP, HBD and HBA values at physiological pH have been calculated with Maestro (Schrödinger Release 2018-4: Maestro, Schrödinger, LLC, New York, NY, 2018).

**Table 1 ijms-20-01402-t001:** Physicochemical property ranges and drug-likeness percentage of flavour molecules. Drug-likeness indices are: 200 < MW < 500 Daltons, −1 < AlogP < 5, HBAs < 5, HBDs < 3, as found for approved DrugBank compounds.

		MW	AlogP	HBAs	HBDs
**Bitter tastants**	Range	27–1524	−8.2–12.6	0–28	0–19
	Drug-like	**67%**	**85%**	**70%**	**80%**
**Sweet tastants ^a^**	Range	92–1287	−6.1–5.1	0–29	0–19
	Drug-like	**27%**	**85%**	**23%**	**20%**
**Umami tastants ^b^**	Range	89–388	−6.6–4.2	0–8	0–6
	Drug-like	**70%**	**54%**	**84%**	**78%**
**Odorants**	Range	44–345	−1.2–4.2	0–4	0–2
	Drug-like	**4%**	**99%**	**100%**	**100%**

Except for bitter compounds, for which all data were retrieved from BitterDB [149,150], physicochemical properties at physiological pH have been calculated with Maestro (Schrödinger Release 2018-4: Maestro, Schrödinger, LLC, New York, NY, 2018). ^a^
Appendix A. ^b^
Appendix A.

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
