# Peer review of "Beyond the Flavour: The Potential Druggability of Chemosensory G Protein-Coupled Receptors"

_ijms, 2019, doi:10.3390/ijms20061402_

Reviewer 1 Report

The manuscript "Beyond the flavor" by Di Pizio et al. tackles the relatively unexplored problem of druggability of chemosensory receptors. In general, I find the manuscript interesting and well-written, worth publishing in IJMS. I only have some minor comments:

'evidence' is considered by the Authors as countable (we can find 'evidences' in the manuscript, e.g. line 117, 283). Although such form can be found from time to time, it is not exactly proper and definitely not elegant. I'd suggest to change it to e.g. 'body of evidence' or 'proofs'.

some paragraphs are too short, and should be connected with neighbouring longer ones, especially since they often share the same thought, e.g. line 56/57, line 161/162

in line 269, we can find a sentence '(...)hydrophobicity has been disputed from long.' I feel something is missing here.

In line 86, comma should be replaced with full-stop or with a connector.

line 333, sentence starts with 'and'

line 364, sentence 'odorants are usually smaller'. I understand what the Authors meant, but I think that it would sound better when completed, e.g. 'smaller than drug-like compounds', 'smaller than other molecules described in the manuscript', or simply 'small'.

in Figure 3, we read 'Manifacture' instead of 'Manufacture'.

In Figure 1 caption (line 50) there is 'class A' instead of 'class C', 

Author Response

Point 1. The manuscript "Beyond the flavor" by Di Pizio et al. tackles the relatively unexplored problem of druggability of chemosensory receptors. In general, I find the manuscript interesting and well-written, worth publishing in IJMS. I only have some minor comments:

Response 1. We thank reviewer 1 for the careful reading of our manuscript and the positive feedback, we tried to address all comments as detailed below and we believe this led to an improved manuscript. We are submitting a version in which all changes are highlighted.

Point 2. 'evidence' is considered by the Authors as countable (we can find 'evidences' in the manuscript, e.g. line 117, 283). Although such form can be found from time to time, it is not exactly proper and definitely not elegant. I'd suggest to change it to e.g. 'body of evidence' or 'proofs'.

Response 2. We have changed the use of ‘evidence’ with a more appropriate wording. Specifically, lines 59, 92, 112, 130, 188, 309 have been modified.  

Point 3. some paragraphs are too short, and should be connected with neighboring longer ones, especially since they often share the same thought, e.g. line 56/57, line 161/162

Response 3. We thank the reviewer for this suggestion. Paragraphs 1 and 2 have been grouped. Paragraphs 4 and 5 (old numbering) have been grouped and reorganized in sub-paragraphs. Therefore, now the manuscript is shaped in a more schematic way, with Introduction, paragraphs 2 and 3 (three sub-paragraphs each) and conclusions.

Point 4. in line 269, we can find a sentence '(...)hydrophobicity has been disputed from long.' I feel something is missing here.

Response 4. Fixed (now line 290)

Point 5. In line 86, comma should be replaced with full-stop or with a connector.

Response 5. Done (now lines 88-90)

Point 6. line 333, sentence starts with 'and'

Response 6. Fixed. Now line 360.

Point 7. line 364, sentence 'odorants are usually smaller'. I understand what the Authors meant, but I think that it would sound better when completed, e.g. 'smaller than drug-like compounds', 'smaller than other molecules described in the manuscript', or simply 'small'.

Response 7. Fixed (now lines 397-398).

Point 8. in Figure 3, we read 'Manifacture' instead of 'Manufacture'.

Response 8. Fixed, thanks for noticing this typo.

Point 9. In Figure 1 caption (line 50) there is 'class A' instead of 'class C', 

Response 9. Done

Reviewer 2 Report

The review by Di Pizio et al. addresses the topic of chemosensory GPCRs, specifically odorant, trace amine-associated, and tastant receptors. These receptors are typically expressed in discrete tissues where they mediate chemosensation, but are also expressed in other tissues, where presumably they have additional functions. The authors review their cognate roles and additional functions to set the stage for targeting these receptors therapeutically. The authors then discuss the possibility of drug discovery at these receptors. Overall this is an interesting review, the topic is somewhat timely, and should be of interest to a broad readership. I only have minor concerns, but they need to be addressed.

1.    Figure 1 is fine. My comment is in regard to the font used for odor, amine odor, etc. The font should be the same as that used for the receptor class below.

2.    Ec-nomotopic is an interesting suggestion to describe the presence of ORs in extra-nasal space. I’m not sure I understand the rationale for trying to invent a word when the term “extra-nasal” is sufficient and already exists.

3.    In the paragraph beginning at line 88 the authors highlight that because ORs are highly expressed in cancer tissue, “there is potential to serve as therapeutic tools”. How exactly? Their upregulation seems beneficial, since their activation promotes anti-cancer effects. Do tumors express OR antagonists? Does the simple presence of unactivated receptors have deleterious effects? Clarifying these questions in the manuscript will help  the reader understand the significance better.

4.    Line 113-115, the authors write: “In addition, there are several evidences that the OR51E2 agonist β-ionone 113 has effects on prostate cancer: it can cause a decreased proliferation but also an increase of the 114 invasiveness of human prostate cancer cells [77-79]. The use of the evidence in the plural form is not accurate and should be corrected here and other places where it is used in the text. Perhaps rephrase the above sentence to: “In addition, there are several lines of evidence that….”

5.    Other parts of the manuscript could benefit from heavy editing.

Author Response

We thank reviewer 2 for the careful reading of our manuscript and the positive feedback, we tried to address all comments as detailed below and we believe this led to an improved manuscript. We are submitting a version in which all changes are highlighted.

Point 1.    Figure 1 is fine. My comment is in regard to the font used for odor, amine odor, etc. The font should be the same as that used for the receptor class below.

Response 1. Fixed

Point 2.    Ec-nomotopic is an interesting suggestion to describe the presence of ORs in extra-nasal space. I’m not sure I understand the rationale for trying to invent a word when the term “extra-nasal” is sufficient and already exists.

Response 2. We need a term that could be used for both extra-nasal and extra-oral expression of ORs and taste receptors, respectively. We re-edited this part to make it clearer (lines 72-85). Anyway, we took the occasion to discuss and suggest a replacement for the use of the term “ectopic”, that is not correct and broadly used even when not really needed (instead of the term “extra-nasal”).

Point 3.    In the paragraph beginning at line 88 the authors highlight that because ORs are highly expressed in cancer tissue, “there is potential to serve as therapeutic tools”. How exactly? Their upregulation seems beneficial, since their activation promotes anti-cancer effects. Do tumors express OR antagonists? Does the simple presence of unactivated receptors have deleterious effects? Clarifying these questions in the manuscript will help the reader understand the significance better.

Response 3. Thank you for addressing this unclear point. The fact that some ORs are highly expressed in cancer first suggested these ORs as diagnostic tools. In addition, when the role of OR in cancer will be clarified, ORs may become also therapeutic targets since we can modulate them with small molecules, and the patient can benefit from the OR-modulated anti-cancer effects. The effects depend on which ligand is used (agonist, partial agonist, inverse agonist, antagonist). Moreover, ORs have been found to be oncotargets and oncosuppressors, therefore their roles and effects are different case by case. Indeed, this is a complex system and further studies are required to understand the complete picture. This is beyond the scope of our review, since we are mainly interested in zooming-in to the possibility to target ORs in cancer. However, to make the picture clearer, the sentence in line 88 (previous version of the manuscript) was moved to the `Ecnomotopic ORs` paragraph when we discuss about ORs in cancer (lines 108-109); and we showed the complexity by discussing the case of OR51E2 and the different effects of agonists on cell proliferation (lines 110-121). Particularly, we added (and refer for more details) to the study from Abaffy et al 2018 https://www.frontiersin.org/articles/10.3389/fonc.2018.00162/full#h5. In this paper, high OR51E2 expression, endogenous ligands’ production and receptor’s activation are analyzed, the role and the potential therapeutic use of OR51E2 in prostate adenocarcinoma are suggested.

Point 4.    Line 113-115, the authors write: “In addition, there are several evidences that the OR51E2 agonist β-ionone 113 has effects on prostate cancer: it can cause a decreased proliferation but also an increase of the 114 invasiveness of human prostate cancer cells [77-79]. The use of the evidence in the plural form is not accurate and should be corrected here and other places where it is used in the text. Perhaps rephrase the above sentence to: “In addition, there are several lines of evidence that….”

Response 4. Thank you, we have fixed this point (lines 59, 92, 112, 130, 188, 309 have been modified).

Point 5.    Other parts of the manuscript could benefit from heavy editing.

Response 5. The revision work has led to a substantial re-editing of the manuscript: the text was rearranged, so that now there are 4 paragraphs: introduction, paragraph 2 about ecnomotopic receptors (with three sub-paragraphs), paragraph 3 about druggability of ecnomotopic receptors (with three sub-paragraphs) and conclusions. Many parts of the manuscript have been re-written.